# Appropriate Method of Administering Vasopressors for Maternal Hypotension Associated with Combined Spinal Epidural Anesthesia in Elective Cesarean Section: Impact on Postnatal Respiratory Support for Newborns

**DOI:** 10.3390/medicina58030403

**Published:** 2022-03-08

**Authors:** Shoichi Magawa, Masafumi Nii, Yosuke Sakakura, Naosuke Enomoto, Sho Takakura, Shintaro Maki, Hiroaki Tanaka, Eiji Kondo, Tomoaki Ikeda

**Affiliations:** 1Department of Obstetrics and Gynecology, Faculty of Medicine, Mie University, Tsu 514-8507, Mie, Japan; doaldosleeping@yahoo.co.jp (M.N.); nao-e@clin.medic.mie-u.ac.jp (N.E.); stakakura813@gmail.com (S.T.); mabochikin519@yahoo.co.jp (S.M.); h_tanaka@med.miyazaki-u.ac.jp (H.T.); eijikon@clin.medic.mie-u.ac.jp (E.K.); t-ikeda@clin.medic.mie-u.ac.jp (T.I.); 2Department of Clinical Anesthesiology, Faculty of Medicine, Mie University, Tsu 514-8507, Mie, Japan; sakakura-ysk@med.mie-u.ac.jp

**Keywords:** cesarean section, combined spinal–epidural anesthesia, maternal hypotension, neonate, non-reassuring fetal status, vasopressor

## Abstract

*Background and Objectives*: Vasopressors are used for treating maternal hypotension. However, the appropriate administration method and effects on newborns have not been reported. We evaluated maternal blood pressure fluctuation and neonatal findings in patients who received continuous vasopressor administration during elective cesarean sections and those who received bolus vasopressor administration upon onset of hypotension. *Materials and Methods*: We retrospectively analyzed the data of 220 patients scheduled for elective cesarean delivery under spinal anesthesia at Mie University Hospital between April 2017 and March 2021. The patients were classified according to the method of vasopressor administration. Maternal information, intraoperative maternal blood pressure fluctuation, and neonatal findings were examined. A multiple regression analysis was performed for the administration of postpartum neonatal respiratory support using maternal background information and other variables related to blood pressure changes as independent variables. *Results*: The Continuous group and the Bolus group were composed of 98 and 122 patients, respectively. No difference was observed in maternal background information between the groups. Significant changes were noted in several blood pressure parameters between both groups. As for neonatal parameters, newborns of Bolus group patients had lower pO_2_, 1 min and 5 min Apgar scores, and required more respiratory support than those of Continuous group patients. In the multiple regression analysis, the groups and maternal post-anesthesia diastolic blood pressure variability were considered explanatory variables. *Conclusions*: Maternal hypotension and the need for neonatal respiratory support associated with anesthesia administration in elective cesarean section may be improved by continuous vasopressor administration upon induction of combined spinal–epidural anesthesia.

## 1. Introduction

Combined spinal–epidural anesthesia (CSE anesthesia) is often used for elective cesarean sections because it is characterized by reduced medication exposure for the fetus compared to general anesthesia. Moreover, it allows the mother to remain conscious during delivery. However, the main side effect of the administration of spinal anesthesia is maternal hypotension, which affects 90% of patients and has been suggested to cause nausea and vomiting. In severe cases, fetal bradycardia and cardiovascular collapse may ensue [1].

Although there are various reports on this issue, studies on the continuous sequence of events involving the mother and newborn are scarce. For example, in a study that compared general anesthesia and CSE anesthesia for elective cesarean section, no difference was observed in the development of transient tachypnea between newborns that received general anesthesia and those that received CSE anesthesia, although maternal blood pressure was not evaluated [2].

Spinal anesthesia has also been reported to be associated with neonatal acidosis, with sustained hypotension, and time to delivery is being implicated with this adverse event [3]. Clinically, vasopressors are commonly administered for maternal hypotension associated with spinal anesthesia. However, there has been no consensus on their appropriate use.

Ephedrine, an alpha- and beta-receptor stimulant, or phenylephrine, an alpha-receptor stimulant, are commonly used vasopressors. Although there is an ongoing debate concerning the better option between these two vasopressors, only a few studies have reported the superiority of phenylephrine over ephedrine [4,5,6].

Moreover, reports describing maternal and neonatal effects of vasopressor administration are few. A study that compared bolus versus continuous administration of phenylephrine in elective cesarean sections performed with limited resources, such as in South Africa, concluded that phenylephrine use contributed to the prevention of maternal hypotension, although there was no detailed evaluation of its effects on newborns [7].

We investigated continuous phenylephrine administration before CSE anesthesia to prevent maternal hypotension and continuous bolus administration of phenylephrine or ephedrine when there was an actual decrease in blood pressure. The primary outcome was a reduction in maternal blood pressure variability, and the secondary outcome was the effect on the newborn. Therefore, we focused on maternal blood pressure variability and maternal background, and we examined the neonatal impact from multiple perspectives. The phenylephrine model and the bolus model, in which no prophylactic hypotensive agents were administered and a bolus dose of phenylephrine or ephedrine was administered when hypotension was observed, were selected to analyze maternal circulatory changes and the effects on the newborn. The effects of these two types of administration on maternal and neonatal conditions were examined considering the maternal background so that a more appropriate anesthesia method could be considered for elective cesarean section, in which the condition of the fetus shortly before the cesarean section is presumed to be stable.

## 2. Materials and Methods

The protocol was approved by the Mie University Clinical Research Ethics Committee in September 2021 (registration number: H2021-181).

The study was not a randomized controlled trial. It is an observational study evaluating the maternal and fetal effects of different uses of vasopressors in elective cesarean sections performed during the study period. We retrospectively analyzed the medical records of 297 patients who were scheduled for elective cesarean delivery under spinal anesthesia during normal working hours at Mie University Hospital between April 2017 and March 2021. The reasons for a cesarean section are pregnancy after myomectomy, previous cesarean section, abnormal placental position, or maternal skeletal abnormalities. The exclusion criteria were patients with maternal complications that could affect the newborn, such as age <18 years, twin pregnancies, known fetal malformations, fetal growth abnormalities, and maternal heart disease and maternal hypertension requiring medical treatment. Patients with missing umbilical artery blood gas data and those with a maternal systolic blood pressure of 90 mmHg that could not be confirmed by anesthesia records before bolus administration of a vasopressor were also excluded.

All patients were subjected to routine monitoring and, at the discretion of the anesthesiologist, received intravenous administration of prehydration, rehydration, or both via a large-bore venous access established in the upper extremity. All patients received CSE anesthesia, using 2 mL of 0.5% marcaine and 10 μg of fentanyl. The height of the block was assessed via cold stimulation with ice before surgery. In cases of continuous administration, 1 mg phenylephrine was adjusted in a 10 mL syringe and administered at 0.3 μg/kg/min as a base dose. For bolus administration, when the maternal systolic blood pressure decreased to <90 mmHg after anesthesia, 0.1 mg of phenylephrine or ephedrine (5 mg) was administered, and, if there was no improvement, additional boluses were administered. Anesthesia for all cases is performed by specialists certified by the Japanese Society of Anesthesiologists, and all anesthesiologists are in agreement on how to manage anesthesia for cesarean section.

Moreover, other medications (vasodilators, atropine, antiemetics, and analgesics) were available, as needed. Data collection continued from the induction of anesthesia until the delivery of the baby. In this study, blood pressure was measured at 1 min intervals, and due to the system, the blood pressure progress was recorded at 2.5 min intervals. The monitoring data were electronically recorded, and the anesthetist described the indications for surgery, the drugs administered, and the amount and method of vasopressors administered. Additionally, maternal background information (age, gestational weeks, number of deliveries, pre-pregnancy body mass index (BMI), and BMI at delivery) and neonatal information (umbilical cord blood gas analysis and Apgar score (AS)) were extracted from the electronic medical records in this study. Data on maternal blood pressure (maximum systolic blood pressure (maxSBP), minimum systolic blood pressure (miniSBP), pulse pressure (difference of SBP), maximum diastolic blood pressure (maxDBP), and minimum diastolic blood pressure (miniDBP), and difference in maximum–minimum diastolic blood pressure (difference of DBP)) were extracted from anesthesia records, which were automatically recorded every few min from the induction of spinal anesthesia until delivery. Information on whether the infant received neonatal respiratory support within 24 h of birth was also extracted from the electronic records. Regarding the type of respiratory support, the lower limit was set at the start of oxygen administration in an infant incubator in the NICU, and intubation management was evaluated. Transient use of oxygen for postnatal resuscitation in the operating room was not included as a form of respiratory support.

### Statistical Analysis

Maternal background information and blood pressure-related parameters after anesthesia were compared between the two groups using the Mann–Whitney U test. The need for neonatal respiratory support and AS at 1 min after birth were assessed using the Chi-square test. A multiple regression analysis was performed, with respiratory support within 24 h after birth as the dependent variable and variables regarding the maternal background and blood pressure change as independent variables. Variables were selected using a stepwise method.

The correlation between individual blood pressure-related parameters and the relationship between blood pressure-related parameters and neonatal respiratory support were evaluated for all patients. Correlation analysis was performed using Spearman’s correlation coefficient, and a simple linear regression analysis was conducted to determine the association of individual parameters with neonatal respiratory support. The percentage of neonates with an AS of less than 7 was determined at 1 and 5 min after birth. Analyses were performed using SPSS 26.0 (version 26; IBM Corporation, Armonk, NY, USA). Statistical significance was set at *p* < 0.05.

## 3. Results

Of the 297 patients who underwent elective cesarean section during the study period, 220 were enrolled in the study, with 98 in the Continuous group and 122 in Bolus group. Altogether, 77 participants were excluded from the analysis. Of the 77 cases, 20 were cases with missing data (no umbilical cord blood gas test or missing blood pressure change information in the anesthesia record), and the remaining 57 were inappropriate cases (multiple pregnancies, pregnancies with maternal or fetal heart disease, pregnancies with maternal respiratory disease, neonates requiring postnatal respiratory support such as congenital diaphragmatic hernia, and pregnancies with hypertension requiring medical treatment).

Hence, we finally analyzed 98 patients in the Continuous group and 122 patients in the Bolus group. The indications for the patients and the analysis process are shown in Figure 1. Patient characteristics are summarized in Table 1. Details of blood pressure-related parameters and the time from spinal anesthesia to delivery are summarized in Table 2. No significant difference in maternal background information was observed between the two groups. For blood pressure-related parameters, group differences in maxSBP (*p* = 0.003) and miniSBP (*p* < 0.001) and differences in SBP (*p* < 0.001), miniDBP (*p* < 0.001), and DBP (*p* < 0.001) were noted. Meanwhile, no group differences in maximum DBP and operative times were noted. The results of the neonatal-related parameters, such as umbilical cord blood gas analysis and AS, are summarized in Table 3. Group differences were noted in the pO_2_ value of umbilical cord blood gas analysis (*p* < 0.001) and in the 1 min and 5 min AS (*p* < 0.001 and *p* = 0.029, respectively). There were no cases of AS values less than 7 at 5 min after birth (AS-5) in both groups. There was a significantly higher number of patients with an AS of less than 7 at 1 min (AS-1) in the Bolus group (16 patients; 13.1%) than in the Continuous group (2 patients; 2.0%) (*p* = 0.003) (Figure 1). The percentage of neonates requiring respiratory support within 24 h was 12.2% (12/98) in the Continuous group and 41.0% (50/122) in the Bolus group (*p* < 0.001). Regression analysis of the association between the need for respiratory support and maternal background factors and parameters related to blood pressure changes after spinal anesthesia are summarized in Table 4. For maternal background factors, the group (Continuous group or Bolus group) (*p* < 0.001) was selected as an explanatory variable, and for blood pressure-related parameters, the difference in DBP (*p* < 0.001) was selected. As an additional analysis, regression analysis of individual blood pressure-related parameters and the need for neonatal respiratory support and the results of correlation analysis between blood pressure-related parameters are shown in Table 5 and Appendix A, respectively.

Blood pressure-related parameters were strongly correlated with one another, and the individual blood pressure-related parameters were associated with the need for respiratory support for newborns upon single regression analysis.

## 4. Discussion

In this study, the Bolus group demonstrated greater maternal blood pressure changes from the administration of spinal anesthesia until delivery than the Continuous group. The newborns of patients in the Bolus group had lower pO_2_ and lower AS-1 min and AS-5 min values at birth than those of patients in the Continuous group. The rate of the requirement of respiratory support within 24 h of birth was higher in the Bolus group than in the Continuous group. Upon multiple regression analysis, respiratory support and maternal parameters were analyzed as the dependent variable and explanatory variables, respectively, and the method of administration of vasopressors and the difference of DBP were selected as significant explanatory variables.

Bolus administration of vasopressors was shown to influence maternal blood pressure changes. It has been mentioned that maternal blood pressure change may impact the results of umbilical artery blood gas analysis. It has also been suggested that spinal anesthesia causes vasodilation, decreases blood pressure and maternal cardiac output, and increases the pulsatility index of the umbilical artery [8]. Therefore, a decrease in fetal blood flow occurs, and, in severe cases, it is associated with fetal acidemia [9,10].

Here, the vasopressor was administered only after hypotension was confirmed. Although hypotension did not persist, the Bolus group demonstrated more severe hypotension in both systole and diastole, suggesting that the improvement in the severity of hypotension by the prophylactic continuous administration of vasopressors reduced the severity of fetal hypocapnia and contributed to the suppression of the pO_2_ decrease.

Regarding low umbilical artery blood pO_2_ upon delivery, it has been shown that in the presence of underlying pathologies, such as fetal growth restriction, pO_2_ in umbilical cord blood is reduced, reflecting impaired oxygen diffusion due to long-term abnormal placental development [11]. Meanwhile, low umbilical artery blood pO_2_ has also been reported to be inadequate in predicting high levels of risk in full-term neonates [12]. In summary, the significance of the decrease in umbilical artery blood gas pO_2_ alone is unclear and likely reflects transient abnormal fetal circulation in the prenatal period [13]. Among the neonatal parameters, in addition to pO_2_ in the cord blood gas analysis, AS was also affected by the method of administration of vasopressors. AS is a direct indicator of neonatal well-being [14], and although it does not strictly reflect the results of cord blood gas analysis [15], it is important to assess these two tests together to comprehensively evaluate the neonate.

Here, AS predominantly decreased in the Bolus group compared to the Continuous group. AS is a neonatal assessment score, and a low score reportedly indicates the requirement of respiratory support. In addition, neonatal mortality is inversely correlated with this score. In recent years, there have been many reports on this index, and there are discussions on the use of AS in clinical practice, considering the problem of prematurity in preterm and low-birth-weight infants and the impact of resuscitation interventions [16,17].

We focused on normal singleton pregnancies at term. There are no reports on differences in AS depending on the anesthesia method, with a focus on cesarean section cases. To the best of our knowledge, this study is the first to mention the possibility that the use of vasopressors associated with CSE anesthesia may have also affected AS. In general, it has been pointed out that 5 min values of the AS are more sensitive than 1 min values in assessing neonatal prognosis, and AS < 7 at 5 min is associated with neurological damage, gastrointestinal and infectious morbidities, and neonatal mortality [17,18,19,20]. Moreover, our findings revealed that the Bolus group had significantly lower AS than the Continuous group, which increased the likelihood of neonates requiring respiratory support.

Regarding the need for respiratory support in neonates, which was the primary endpoint of this study, the frequency of need for respiratory support increased predominantly in the Bolus group. Although there are no clear criteria for initiating respiratory support in neonates, it is often used as first-line treatment following non-improvement of tachypnea and work of breathing. This study’s results indicate an association between the use of vasopressors during CSE anesthesia and the difference in DBP after CSE anesthesia in the background of increased requirement for respiratory support in newborns. Similarly, blood pressure-related parameters following CSE anesthesia were correlated with each other, suggesting that blood pressure changes after CSE anesthesia are related to each other. The decrease in blood pressure associated with CSE anesthesia first decreased the activity of fetal circulation. The greater the drop in blood pressure, the worse the fetal circulation. This transient deterioration affects umbilical cord blood gas analysis in the form of pO_2_. This also affects the decreased AS at birth. The final scenario is that medical intervention in the form of respiratory support is necessary at birth.

This study’s limitation is that the criteria for the initiation of respiratory support in neonates depended on the discretion of the neonatologist, and there are no definite criteria for the initiation of oxygen therapy. In most cases, respiratory support was initiated due to persistent neonatal tachypnea or an insufficient increase in SpO_2_. Another limitation is the lack of uniformity in the type of pressors administered by bolus upon onset of maternal hypotension. In this study, we only excluded cases with a history of medication that directly affects blood pressure, and we were not able to examine cases with complications that could potentially affect blood pressure. The timing bias of the cases included in the study must also be described. Although the anesthesia management methods, laboratory equipment, and perinatal management system in the study period were consistent, more cases were recruited to the Bolus group in the first half of the study period, and more cases were recruited to the Continuous group in the latter part.

## 5. Conclusions

Therefore, maternal hypotension and respiratory support for newborns associated with CSE anesthesia in elective cesarean sections may be improved by administering vasopressors continuously with the induction of CSE anesthesia. In the future, we will conduct a prospective study involving the continuous administration of vasopressors to evaluate the appropriate dosage of vasopressor drugs and the risk of maternal hypotension.

## Figures and Tables

**Figure 1 medicina-58-00403-f001:**
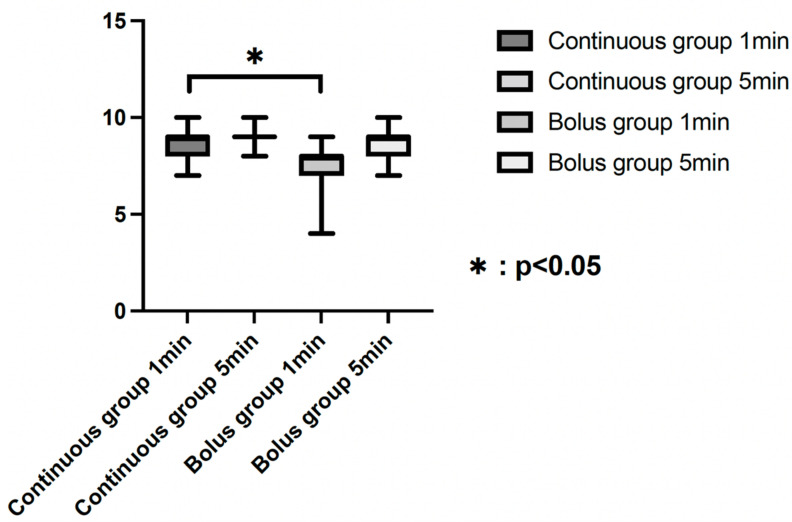
APGAR score (1 min/5 min) in the Continuous group and the Bolus group.

**Table 1 medicina-58-00403-t001:** Background of the Continuous group and the Bolus group.

	Continuous Group	Bolus Group	*p* Score
age	34.58 ± 5.40	34.27 ± 4.93	0.665
para	1.85 ± 0.93	1.97 ± 0.74	0.296
gestational weeks	38.30 ± 0.92	38.27 ± 0.62	0.781
BMI before pregnancy	23.04 ± 4.53	23.16 ± 4.98	0.855
BMI on delivery	26.55 ± 4.47	26.87 ± 5.13	0.627

BMI: body mass index.

**Table 2 medicina-58-00403-t002:** The details of blood pressure-related parameters and the time from spinal anesthesia to delivery.

	Continuous Group	Bolus Group	*p* Score
Max SBP	135.7 ± 14.1	143.3 ± 23.5	0.003
Mini SBP	109.3 ± 12.6	85.3 ± 12.2	<0.001
SBP deference	26.3 ± 11.9	58.0 ± 24.5	<0.001
Max DBP	80.8 ± 11.2	83.2 ± 14.9	0.174
Mini DBP	57.4 ± 13.7	39.8 ± 10.8	<0.001
DBP deference	23.4 ± 10.6	43.4 ± 17.9	<0.001
time	25.9 ± 6.5	27.1 ± 5.4	0.160

time: the time from the start of anesthesia to the baby delivery.

**Table 3 medicina-58-00403-t003:** The results of the neonatal-related parameters.

	Continuous Group	Bolus Group	*p* Score
pH	7.27 ± 0.05	7.25 ± 0.07	0.089
pCO_2_	49.1 ± 8.9	49.9 ± 8.7	0.499
pO_2_	20.3 ± 7.2	14.5 ± 5.7	<0.001
BE	−4.3 ± 2.6	−4.2 ± 2.3	0.411
AS 1 min	7.9 ± 0.5	7.3 ± 0.07	<0.001
AS 5 min	8.8 ± 0.4	8.7 ± 0.5	0.029
AS 1 min < 7 (cases)	2 cases (2.0%)	16 cases (13.1%)	0.003

**Table 4 medicina-58-00403-t004:** The results of the regression analysis conducted to evaluate the association between the need for respiratory support and maternal background factors and parameters related to blood pressure changes after spinal anesthesia.

Background Associated with Respiratory Support		
	β	*p*
Continuous/Bolus group	0.317	<0.001
F score	24.438	
R_2_	0.101	
adjusted R_2_	0.970	
Blood pressure associated with respiratory support		
	β	*p*
DBP deference	0.254	<0.001
F score	14.993	
R_2_	0.064	
adjusted R_2_	0.060	

**Table 5 medicina-58-00403-t005:** Regression analysis of individual blood pressure-related parameters and the need for neonatal respiratory support.

MaxSBP Associated with Respiratory Support		
	β	*p*
maxSBP	0.199	0.005
F score	7.944	
R_2_	0.035	
adjusted R_2_	0.031	
miniSBP associated with respiratory support		
	β	*p*
miniSBP	0.621	<0.001
F score	137.029	
R_2_	0.386	
adjusted R_2_	0.383	
SBP deference associated with respiratory support		
	β	*p*
SBP deference	0.317	<0.001
F score	24.438	
R_2_	0.101	
adjusted R_2_	0.970	
maxDBP associated with respiratory support		
	β	*p*
maxDBP	0.172	0.010
F score	6.664	
R_2_	0.030	
adjusted R_2_	0.025	
miniDBP associated with respiratory support		
	β	*p*
miniDBP	−0.154	0.022
F score	5.310	
R_2_	0.024	
adjusted R_2_	0.019	
DBP deference associated with respiratory support		
	β	*p*
DBP deference	0.254	<0.001
F score	14.993	
R_2_	0.064	
adjusted R_2_	0.060	

## Data Availability

Not applicable.

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
