# Peer review of "Appropriate Method of Administering Vasopressors for Maternal Hypotension Associated with Combined Spinal Epidural Anesthesia in Elective Cesarean Section: Impact on Postnatal Respiratory Support for Newborns"

_medicina, 2022, doi:10.3390/medicina58030403_

Round 1
Reviewer 1 Report
Dr. Magawa et al presented an interesting manuscript to explore the impact of administration of vasopressors on mother and neonatal during elective Caesarean section. The study was conducted by analyzing intra-op data retrospectively and 220 samples of umbilical artery blood gas certainly deserve recognition. The study was in consistent with other publications that even short episode of hypotension may cause harm to our patients.
There are several key elements missing in the paper.
- There is no clear stated hypothesis in the manuscript. Authors wanted to investigate the continuous infusion of phenylephrine and bolus of vasopressors, but there is no hypothesis, nor primary and secondary outcomes listed when they designed the study. There is no power calculation of number of patients needed for the study.
- There are several red flags of their methods and study design. There is a high percentage of exclusion in retrospective data. There are ill defined continuous infusion and bolus groups. The infusion dosage of starting 0.3μg is mis-representing as infusion should be set as μg/min or μg/kg/min.
- Outcomes of the study are not clear. What does respiratory for neonatal mean? It counts for brand spectrum of supplement of oxygen to intubation.
- Baseline demographic data miss several key comorbidities that can influence patient response to vasoactive agents such as hypertension, diabetes, pre-eclampsia as well as home or hospital administrated medications.
- Anesthesia provider input and experience regarding selection of infusion versus bolus was not discussed. This can create selection bias between the two groups. The unusual high percentage of low Apgar score in bolus group gave the indication of bias.
- There are several variables not controlled in the study. It seems that bolus group patients had bigger swing of blood pressure and more severe hypotension, close to 30 mmHg of DBP in some patients. However, the interval of blood pressure monitoring was not specified. Could it be that infusion group patients were better monitored? Any difference of experience of anesthesia providers? Any other home or hospital medications that can interfere with hemodynamic response?
- Table 3, AS 1 min <7 is wrongly calculated and inconsistent with what authors stated in the draft (line 148-149)
- Several choice of words can cause confusion such as continuous bolus, group B and group C (where is group A). Certain paragraph is difficult to understand (line 132-134)
Author Response
Thank you very much for your thoughtful comments on our paper.
We would like to respond to your comments and clarify the corrections we made.
- There is no clear stated hypothesis in the manuscript. Authors wanted to investigate the continuous infusion of phenylephrine and bolus of vasopressors, but there is no hypothesis, nor primary and secondary outcomes listed when they designed the study. There is no power calculation of number of patients needed for the study.
Response
Thank you for your suggestion. The hypothesis of this study is that continuous phenylephrine infusion will reduce post-anesthesia maternal blood pressure variability compared to bolus infusion, and this will contribute to neonatal outcomes. The primary outcome was a reduction in maternal blood pressure variability, and the secondary outcome was the effect on the newborn. Therefore, we focused on maternal blood pressure variability, maternal background, and examined the neonatal impact from multiple perspectives. For this reason, we apologize for the lack of coherence in the text and have added a revised text to the Introduction. Regarding the power to detect the number of patients, given the exploratory nature of this study, which is a retrospective observational study, we decided to evaluate the cases during the study period. (L66-69)
- There are several red flags of their methods and study design. There is a high percentage of exclusion in retrospective data. There are ill defined continuous infusion and bolus groups. The infusion dosage of starting 0.3μg is mis-representing as infusion should be set as μg/min or μg/kg/min.
Response
Thank you for your kind remarks.
Since our institution is a university hospital with a large number of high-risk pregnancies, especially pregnancies complicated by cardiac disease, we had a large number of excluded cases.
Of the 77 cases, 20 were cases with missing data (no umbilical cord blood gas test or missing blood pressure change information in the anesthesia record), and the remaining 57 were inappropriate cases (multiple pregnancies, pregnancies with maternal or fetal heart disease, pregnancies with maternal respiratory disease, pregnancies requiring postnatal respiratory support such as congenital diaphragmatic hernia, and pregnancies with hypertension requiring medical treatment). We have added this information to the Results section. (L147-152)
The definitions of the continuous and bolus groups are explained below. In the continuous group, phenylephrine is administered continuously at a dose of 0.3 gammas. In the bolus group, the bolus dose of the pressure-boosting drug is the amount described in the article when hypotension occurs. We have made an error in describing the dosage, and it should have been 0.3 μg/kg/min since it is 0.3 gamma. We have corrected it. (L98)
- Outcomes of the study are not clear. What does respiratory for neonatal mean? It counts for brand spectrum of supplement of oxygen to intubation.
Response
Thank you for pointing this out.
We apologize for the difficult-to-understand description. In this study, we are evaluating whether oxygen is administered to newborns in situations other than resuscitation. The purpose of this study is to evaluate the use of oxygen in cases where some kind of respiratory abnormality is expected after the resuscitation period and oxygen administration is required. This protocol and phrase were chosen to more accurately assess respiratory abnormalities in newborns, as intubation is rarely required in elective cesarean sections in normal pregnancies.
- Baseline demographic data miss several key comorbidities that can influence patient response to vasoactive agents such as hypertension, diabetes, pre-eclampsia as well as home or hospital administrated medications.
Response
Thank you for your comments.
In this study, hypertensive cases requiring medical management and those directly related to cardiac dynamics were excluded. We have added the above information. (L88)
Diabetes mellitus was observed in 3 patients in the continuous group and 5 patients in the bolus group. Other diseases that may potentially affect blood pressure also need to be evaluated.
However, in this study, we only excluded cases of pregnancy complicated by hypertension requiring medical treatment, which were thought to directly affect blood pressure. For this reason, we have added the following sentence to the Limitation: "In this study, we only evaluated cases with a history of medication that directly affects blood pressure, and we were not able to examine cases with complications that could potentially affect blood pressure.” (L267-269)
- Anesthesia provider input and experience regarding selection of infusion versus bolus was not discussed. This can create selection bias between the two groups. The unusual high percentage of low Apgar score in bolus group gave the indication of bias.
Response
There are only two anesthesia supervisors at the facility where this study was conducted, both of whom are board certified anesthesiologists. The anesthesia management system was maintained during the study period, and it is assumed that there is unlikely to be any difference in anesthesia methods between the two groups. In the first half of the study period, there were many cases of bolus administration, and then the number of cases of continuous phenylephrine administration increased as a result of review by the supervising physicians with reference to reports on the efficacy of continuous phenylephrine administration (Bishop et al., 2017; Xiao et al., 2020). For these reasons, we believe that there is little selection bias by anesthesiologists. In addition, there has been no change in the management methods in obstetrics and gynecology and neonatology during this period, and no change in the measurement equipment. As for the experience of the anesthesiologist, we have added the information in Methods section. (L101-103)
- There are several variables not controlled in the study. It seems that bolus group patients had bigger swing of blood pressure and more severe hypotension, close to 30 mmHg of DBP in some patients. However, the interval of blood pressure monitoring was not specified. Could it be that infusion group patients were better monitored? Any difference of experience of anesthesia providers? Any other home or hospital medications that can interfere with hemodynamic response?
Response
Thank you for pointing this out.
In this study, blood pressure was measured at 1-minute intervals, and due to the system, the blood pressure progress was recorded at 2.5-minute intervals. Because of this system, phenylephrine was administered to some patients even when their systolic blood pressure did not reach 90 mmHg or lower in the records. We excluded these cases because the records did not reflect the hypotension detected by the blood pressure measurement at 1-minute intervals. We have added a note about the interval of blood pressure measurement. (L106-107)
Also, I have responded above regarding the experience of the anesthesiologist and the patient's medication.
- Table 3, AS 1 min <7 is wrongly calculated and inconsistent with what authors stated in the draft (line 148-149)
Response
Thank you for pointing this out. It was an error in the data entry and we fixed it.
- Several choice of words can cause confusion such as continuous bolus, group B and group C (where is group A). Certain paragraph is difficult to understand (line 132-134)
Response
Thank you for pointing this out. We did not take it into consideration, and instead of abbreviating it, we described it as Bolus group and Continuous group. In addition, as you pointed out, we described the contents of the excluded cases, and we apologize for the difficult-to-understand text. I have corrected it as I received it in a previous question.(L147-152)
REFERENCE
Bishop DG, Cairns C, Grobbelaar M & Rodseth RN. (2017). Prophylactic Phenylephrine Infusions to Reduce Severe Spinal Anesthesia Hypotension During Cesarean Delivery in a Resource-Constrained Environment. Anesth Analg 125, 904-906.
Xiao F, Shen B, Xu WP, Feng Y, Ngan Kee WD & Chen XZ. (2020). Dose-Response Study of 4 Weight-Based Phenylephrine Infusion Regimens for Preventing Hypotension During Cesarean Delivery Under Combined Spinal-Epidural Anesthesia. Anesth Analg 130, 187-193.
Reviewer 2 Report
From my point of view the article is interesting.
The introduction is relevant. At the aim I will explain in more details why it was chosen the elective cesarean section.
The methodology is clear.
At the result I will suggest to add maybe a comparative graphic for APGAR Scores because it may be easy to see the differences than in table.
My question is about BMI. How it is greater before pregnancy that at the delivery time? Do you want to say before delivery and at the delivery time?
Discussion section is interesting but I will avoid phrases such as.. there have been no reports. I suggest...according to our knowledge, there are no reports...
Author Response
Thank you very much for your thoughtful comments on our paper.
We would like to respond to your comments and clarify the corrections we made.
- The introduction is relevant. At the aim I will explain in more details why it was chosen the elective cesarean section.
Thank you for your suggestion. The reasons for a cesarean section are pregnancy after myomectomy, previous cesarean section, abnormal placental position or maternal skeletal abnormalities. We will add this information to the Methods section. (L83-85)
- The methodology is clear.
Thank you very much. I've added the above text you pointed out.
- At the result I will suggest to add maybe a comparative graphic for APGAR Scores because it may be easy to see the differences than in table.
We have created the graph as you suggested. In order to evaluate the statistical significance of the AS 1min values, we used the chi-square test to evaluate the difference between the two groups. This has been added to the section on statistical evaluation and results. (L128)
- My question is about BMI. How it is greater before pregnancy that at the delivery time? Do you want to say before delivery and at the delivery time?
First of all, We apologize for and correct the reversal of the BMI in the Table.
Before pregnancy and delivery time strictly describes BMI at non-pregnancy and just before delivery. And regarding before pregnancy and delivery time BMI, previous reports have shown that maternal BMI before pregnancy (Richardson et al., 2017) or during pregnancy or delivery time (Smid et al., 2016; Liljestrom et al., 2018) affects neonatal outcomes. So, in this study, we evaluated before pregnancy and delivery time BMI between groups.
- Discussion section is interesting but I will avoid phrases such as.. there have been no reports. I suggest...according to our knowledge, there are no reports...
Thank you for your kind remarks. I have corrected them.
REFERENCE
Smid MC, Vladutiu CJ, Dotters-Katz SK, Manuck TA, Boggess KA & Stamilio DM. (2016). Maternal Super Obesity and Neonatal Morbidity after Term Cesarean Delivery. Am J Perinatol 33, 1198-1204.
Liljestrom L, Wikstrom AK, Agren J & Jonsson M. (2018). Antepartum risk factors for moderate to severe neonatal hypoxic ischemic encephalopathy: a Swedish national cohort study. Acta Obstet Gynecol Scand 97, 615-623.
Round 2
Reviewer 1 Report
Thank authors response to our questions.
There are still areas not addressed
- The authors added one more exclusion criteria to their study, "maternal hypertension requiring medical treatment". However, it does not address the selection bias between the two study groups. There are no table of demographic data of their patients beside age such as renal problems, maybe pre-eclampsia without medicine etc.
- Authors did admit there were patients among their study patients. Do they include in their multi-variable regression analysis?
- The outcomes of study is still not clear defined. The primary outcome by authors is blood pressure variability. Is it cohort blood pressure variability or individual patient blood pressure variability during delivery? How can cohort blood pressure variability has impact on neonatal outcomes? The secondary outcome is impact on newborn. Again, need to be clearly define.
- The concern of experience of anesthesia providers is not fully addressed. As authors stated, their hospital is a large academic center. There are only two supervising anesthesiologists which means there would be trainee involved in those patient care. In addition, the two methods of vasopressor administration is heavy on bolus at the early phase of the study and more patients received infusion at the late phase of the study, which is another bias.
- There are still discrepancies in the manuscript that require authors attention. For example, the conclusion at abstract line 25-27 is total opposite of their conclusion and discussion line 198-202.
Author Response
Thank you for your careful review of our work. We would like to reply to the comments and make corrections.
・The authors added one more exclusion criteria to their study, "maternal hypertension requiring medical treatment". However, it does not address the selection bias between the two study groups. There are no table of demographic data of their patients beside age such as renal problems, maybe pre-eclampsia without medicine etc.
Reply
Thank you for your comments.
It is true that in this study, patients who underwent elective cesarean section during the study period were grouped according to the anesthesia method, and potential selection bias cannot be denied. The possibility of selective bias has been added to Limitation (L:269-273). At our institution, patients with hypertension during pregnancy are treated with magnesium sulfate under hospitalization. There were no cases of elevated blood pressure or pre-eclampsia during pregnancy in the patients recruited for this study. As for renal dysfunction, there were no cases of chronic renal disease in the coexisting cases, and Cre was less than 1.0 mg/dl in all cases in the preoperative test for cesarean section.
・Authors did admit there were patients among their study patients. Do they include in their multi-variable regression analysis?
Reply
As you pointed out, a small number of cases were found to have background diseases that may not be directly related to blood pressure severely, and these cases were included in the regression analysis. Regarding complications in the recruited cases, the Continuous Group showed 3 cases of diabetes, 4 cases of depression, and 3 cases of hypothyroidism. In contrast, the Bolus Group showed 4 cases of diabetes, 2 cases of depression, 1 case of hyperthyroidism, and 2 cases of hypothyroidism. Once again, we would like to add that we have not seen abnormally high blood pressure levels during pregnancy in all cases.
・The outcomes of study is still not clear defined. The primary outcome by authors is blood pressure variability. Is it cohort blood pressure variability or individual patient blood pressure variability during delivery? How can cohort blood pressure variability has impact on neonatal outcomes? The secondary outcome is impact on newborn. Again, need to be clearly define.
Reply
Thank you for pointing this out.
Originally, we planned this study because we had the impression that there was a tendency that there was a large disparity in the number of cases requiring respiratory support (in a broad sense) for the newborn even in elective cesarean sections in normal pregnancies. Although previous literature has shown that anesthesia prior to cesarean section lowers maternal blood pressure, there have been no reports that have continuously evaluated the appropriate anesthesia method and the associated effects on the newborn. As described below, we cannot deny the fact that there is a timing bias, but we used two different patterns of vasopressor use for cesarean section, and we needed to confirm whether this difference contributed to whether the neonate required respiratory support or not. So, the primary endpoint is the change in maternal blood pressure due to the difference in the method of use of the vasopressor, which is the variation in cohort blood pressure. In the Bolus Group, where blood pressure variability was greater, it had an impact on the percentage of neonates requiring respiratory support and on umbilical cord blood gas pO2.
However, as you pointed out, the description in the paper is difficult to understand, so we have clearly stated that the primary endpoint is "blood pressure variability by group" and the secondary endpoint is "percentage of neonates requiring respiratory support and laboratory results that affect this".(L:66-71)
・The concern of experience of anesthesia providers is not fully addressed. As authors stated, their hospital is a large academic center. There are only two supervising anesthesiologists which means there would be trainee involved in those patient care. In addition, the two methods of vasopressor administration is heavy on bolus at the early phase of the study and more patients received infusion at the late phase of the study, which is another bias.
Reply
Anesthesia in cesarean section, especially from CSE to delivery of the baby, is a time when the maternal circulation changes significantly, and anesthesia management should be especially careful. At our hospital, this time period is always directly supervised by a supervising anesthetist who is a specialist certified by the Japanese Society of Anesthesiologists. Therefore, although there are times when trainees are involved in care, this is under the supervision of a specialist, and we believe that the impact is minimal. In addition, the timing of elective cesarean section is coordinated with other surgeries that are held on the same day, so that a specialist is always available to monitor the anesthesia. As you pointed out, the anesthesia supervisor, laboratory equipment, and obstetric management system are consistent, but timing bias cannot be denied. As mentioned above, we have added this information to Limitation.
・There are still discrepancies in the manuscript that require authors attention. For example, the conclusion at abstract line 25-27 is total opposite of their conclusion and discussion line 198-202.
Reply
Thank you for pointing out the error in the abstract. We have corrected it to "low pO2 and Apgar scores at 1 and 5 minutes. (L:26)